# SVM Performance for Predicting the Effect of Horizontal Screen Diameters on the Hydraulic Parameters of a Vertical Drop

Rasoul Daneshfaraz [1] , Ehsan Aminvash [1] , Amir Ghaderi [2] , John Abraham [3,*]
and Mohammad Bagherzadeh [4]

1   Department of Civil Engineering, Faculty of Engineering, University of Maragheh,
    Maragheh 8311155181, Iran; daneshfaraz@maragheh.ac.ir (R.D.); ehsan.aminvash@stu.maragheh.ac.ir (E.A.)
2   Department of Civil Engineering, Faculty of Engineering, University of Zanjan, Zanjan 537138791, Iran;
    amir_ghaderi@znu.ac.ir
3   School of Engineering, University of St. Thomas, St. Paul, MN 55105, USA
4   Department of Civil Engineering, Faculty of engineering, Urmia University, Urmia 5756151818, Iran;
    m.bagherzadeh@urmia.ac.ir
*   Correspondence: jpabraham@stthomas.edu; Tel.: +1-612-963-2169

**Abstract:** The present study investigated the application of support vector machine algorithms for predicting hydraulic parameters of a vertical drop equipped with horizontal screens. The study incorporated varying sizes of a rectangular channel. Horizontal screens, in addition to being able to dissipate the destructive energy of the flow, cause turbulence. The turbulence in turn supplies oxygen to the system through the promotion of air–water mixing. To achieve the objectives of the present study, 164 experiments were analyzed under the same experimental conditions using a support vector machine. The approach utilized dimensionless terms that included scenario 1: the relative energy consumption and scenario 2: the relative pool depth. The performance of the models was evaluated with statistical criteria (RMSE, $R^2$ and KGE) and the best model was introduced for each of the parameters. RMSE is the root mean square error, $R^2$ is the correlation coefficient and KGE is the Kling–Gupta criterion. The results of the support vector machine showed that for the first scenario, the third combination with $R^2$ = 0.991, RMSE = 0.00565 and KGE = 0.998 for the training mode and $R^2$ = 0.991, RMSE = 0.00489 and KGE = 0.991 for the testing mode were optimal. For the second scenario, the third combination with $R^2$ = 0.988, RMSE = 0.0395 and KGE = 0.998 for the training mode and $R^2$ = 0.988, RMSE = 0.0389 and KGE = 0.993 for the testing mode were selected. Finally, a sensitivity analysis was performed that showed that the $y_c/H$ and $D/H$ parameters are the most effective parameters for predicting relative energy dissipation and relative pool depth, respectively.

**Keywords:** relative energy dissipation; relative pool depth; support vector machine; vertical drop; horizontal screen





## 1. Introduction

Downstream energy dissipation is inevitable in supercritical flow with hydraulic structures. In order to prevent erosion and degradation of the downstream channel, energy dissipation strategies must be employed. Horizontal screens, in addition to dissipating the kinetic energy of the stream, introduce a large amount of air into the system via air–water mixing downstream of the vertical drops. On the other hand, the use of screens to dissipate the flow energy does not damage the environment and can even be used as a garbage collector to prevent waste from continuing in the water system. Nowadays, vertical and horizontal screens have been proposed as an energy consuming structure; they cause the destruction of energy by both promoting air–water mixing and increasing turbulence in the flow. In order to reduce the volume of earthworks and control the design slope in irrigation and drainage canals, vertical drops are usually used. Due to the turbulence created behind the landing jet, this leads to greater energy consumption than with other

methods. Rajaratnam and Chamani [1] examined the energy dissipation of vertical drops and provided a relation for determining the energy reduction. They found that the angle of the jet varies with the flow conditions and the pool depth.

The effect of using square-shaped stairs attached to a drop wall was investigated by Esen et al. [2]. Their results showed that compared to a simple vertical drop, the presence of a step increases the energy dissipation and that increasing its height increases the dissipation. In Huang et al. [3], vertical drops were studied by considering the positive slope in the lower bed of the drop. They provided relationships for estimating the length and force on the inclination. Daneshfaraz et al. [4] reported that a horizontal screen significantly reduces turbulence length and residual energy.

The application of horizontal screens at the brink of a drop, with and without a downstream rough bed, could be a suitable alternative for a stilling basin. Rajaratnam and Hurtig [5] used vertical screens as an energy dissipating structure; they showed that the energy dissipation generated by screens is much higher than for the classical hydraulic jump. Subsequently, Çakir [6] reported on the lack of an effect of screen thickness on energy dissipation. Buzkuş et al. [7] showed that vertical screens with a porosity of 40% have the greatest effect on energy dissipation. Mahmoud et al. [8] studied vertical screens with a square aperture and found that such screens increase energy dissipation. Sadeghfam et al. [9] reported the effect of double screens on flow energy consumption in their study. Daneshfaraz et al. [10] showed that the use of blocks with vertical screens increases energy dissipation compared to the non-block mode. Daneshfaraz et al. [11] examined flow energy dissipation with vertical screens in an erodible bed and showed that the minimum scour depth occurred in the mesh plate with 50% porosity. A mesh plate with 40% provided greater energy dissipation than the 50% screen. Mansouri and Ziaei [12] numerically studied the effect of a vertical drop positioned to a downstream adverse slope. Results showed that the k-ε turbulence was in better agreement with the experimental data than other turbulence models.

In recent years, researchers have used new methods for analyzing hydraulic performance. These methods include artificial neural networks (ANNs), gene expression programming (GEP), the genetic algorithm (GA), adaptive neuro-fuzzy inference system (ANFIS), and support vector machine (SVM) methods. So far, relatively extensive research has been conducted using the above methods, including the following:

Arffin et al. [13] used the ANN model and linear regression to predict the amount of sediment in water flows. They were able to establish a relationship between four parameters that affect the amount of sediment and sediment density using the two methods. Alp and Cigizoglu [14] used two types of artificial neural networks Feed Forward Back-Propagation (FFBP) and Radial Basis Functions (RBF) and compared the results with a multiple linear regression. They concluded that the neural network provided more accurate results compared to linear regression. Goel and Pal [15] used field and laboratory data to investigate the potential of SVM for predicting scour depth and showed that changes in flow conditions, geometry and substrate materials affect scour depth. Roshangar et al. [16] evaluated the efficiency of SVM for predicting hydraulic jump parameters in a sudden divergence. The results showed that the relative energy dissipation and the ratio of conjugate depths with the Froude number had the best performance predictive performance. Sadeghfam et al. [17] studied the scouring of supercritical current jets upstream of lattice plates in experiments using artificial intelligence methods. In that research, Sugeno fuzzy logic (SFL), neuro-fuzzy (NF) and support vector machine (SVM) methods were used.

Daneshfaraz et al. [18] investigated a vertical drop equipped with double screens using SVM. The results showed that the parameters of the vertical drop correspond very well with the output results of the support vector machine. Norouzi et al. [19] investigated the discharge coefficient of trapezoidal labyrinth weirs using ANN and SVM. The results showed that both methods had good accuracy for estimating discharge coefficients. Alizadeh et al. [20] predicted longitudinal dispersion coefficients in natural rivers using

a cluster-based Bayesian network. Their results showed that a dimensionless bayesian network (BN) model resulted in a 30% reduction of the root mean square error. The accuracy criterion increased from 70 to 83% by performing clustering analysis based on the BN model.

One of the parameters that can be considered in the design and operation of systems with horizontal screens is the aperture diameter of these screens. The effect of an aperture diameter with fixed porosity has not been investigated using artificial intelligence methods such as the support vector machine method. Considering the effect of diameter size of screens on flow hydraulics has not been investigated so far; thus, we consider three different diameters of a constant porosity screen and the effect of this parameter on vertical hydraulic parameters is investigated using SVM.

## 2. Materials and Methods

### 2.1. Experimental Set-Up

One of the objectives of the present study is the application of the SVM algorithm for predicting the effect of horizontal screens on hydraulic characteristics. Experiments were performed in a laboratory flume that is 5 m in length, 0.3 m in width and 0.45 m in height with a zero degree slope. Note that this laboratory flume is similar to previous work done by Ghaderi et al. [21,22] and Daneshfaraz et al. [23]. Transparent walls and floors were used to enable visualization of flow. A vertical drop with a height of 15 cm and a length of 1.20 m was made using glass boxes. The inlet flow to the flume was provided by two pumps, each with a capacity of 450 L per min. The flow rate was read using rotameters installed on the pumps with a relative error of $\pm2\%$. A polyethylene sheet with a thickness of 0.01 m and planar dimensions of $0.7 \times 0.3$ m with zigzag arrangements of holes with diameters of 0.01, 0.02 and 0.03 m was used to make the mesh plates (see, for more details, [4]). Figure 1 shows a schematic of a laboratory flume and the equipment installed on it. In order to measure the depth of flow, a point gage with an error of $\pm1$ mm was used. In Figure 2, the side and top views of the vertical drop are shown, and the range of measured parameters is presented in Table 1.

**Table 1.** Variation range of parameters in the present study.

| Percentage of Screens Porosity | D/H | Parameters | | | | |
|---|---|---|---|---|---|---|
| | | $Q$ (l/s) | $y_0$ (cm) | $y_1$ (cm) | $y_c$ (cm) | $y_p$ (cm) |
| 50% | 0.067 | | | 2.6–7.1 | | 3.7–10.3 |
| | 0.133 | 2.5–13.5 | 4.45–6.5 | 2.8–7.0 | 1.92–5.86 | 3.5–9.8 |
| | 0.2 | | | 2.8–7.1 | | 3.2–9.8 |

To evaluate the performance of the support vector machine (SVM) for predicting the relative energy dissipation ($\Delta E/E_0$) and the relative pool depth ($y_p/H$), a total of 164 datasets were obtained. The first 82 datasets are related to the relative energy consumption and the remaining datasets are related to the relative pool depth. Three different relative diameter sizes were used, namely 0.067, 0.133 and 0.2.

### 2.2. Dimensional Analysis

According to Figure 1 and by considering the geometric and hydraulic parameters, the effective parameters can be written as Equation (1):

$$f_1(q, \rho, \mu, g, B, H, \varepsilon, y_c, y_0, D) = 0 \qquad (1)$$

Here, $q$ is flow rate per unit width, $\rho$ is the density of water, $\mu$ is dynamic viscosity, $g$ is gravitational acceleration, $B$ is the channel width, $H$ is drop height, $\varepsilon$ is screen porosity ratio, $y_c$ is critical depth, $y_0$ is the upstream depth of the drop and $D$ is the pore diameter of

the screens. Using the π-Buckingham dimensional analysis method, the relative energy dissipation and relative pool depth were obtained as Equation (2):

$$\frac{\Delta E}{E_0}, \frac{y_p}{H} = f_2(Fr_0, \varepsilon, \mathrm{Re}_0, \frac{y_c}{H}, \frac{D}{H}) \tag{2}$$

In the above relations, $Re_0$ is the Reynolds number upstream of the drop and $Fr_0$ is the Froude number upstream of the drop. Given that the upstream Reynolds number is in the range 8333–44,444, the flow is Completely turbulent and viscous effects can be ignored. Additionally, the porosity of the screens is constant and equal to 50%, so the effect of porosity variation can be ignored. Furthermore, upstream of the drop, there is subcritical flow and the relative pool depth is independent of the upstream Froude number, so the upstream Froude number can be eliminated [24–27]. Consequently, Equation 2 can be rewritten as Equations (3) and (4):

$$\frac{\Delta E}{E_0} = f_3(Fr_0, \frac{y_c}{H}, \frac{D}{H}) \tag{3}$$

$$\frac{y_p}{H} = f_4(\frac{y_c}{H}, \frac{D}{H}) \tag{4}$$

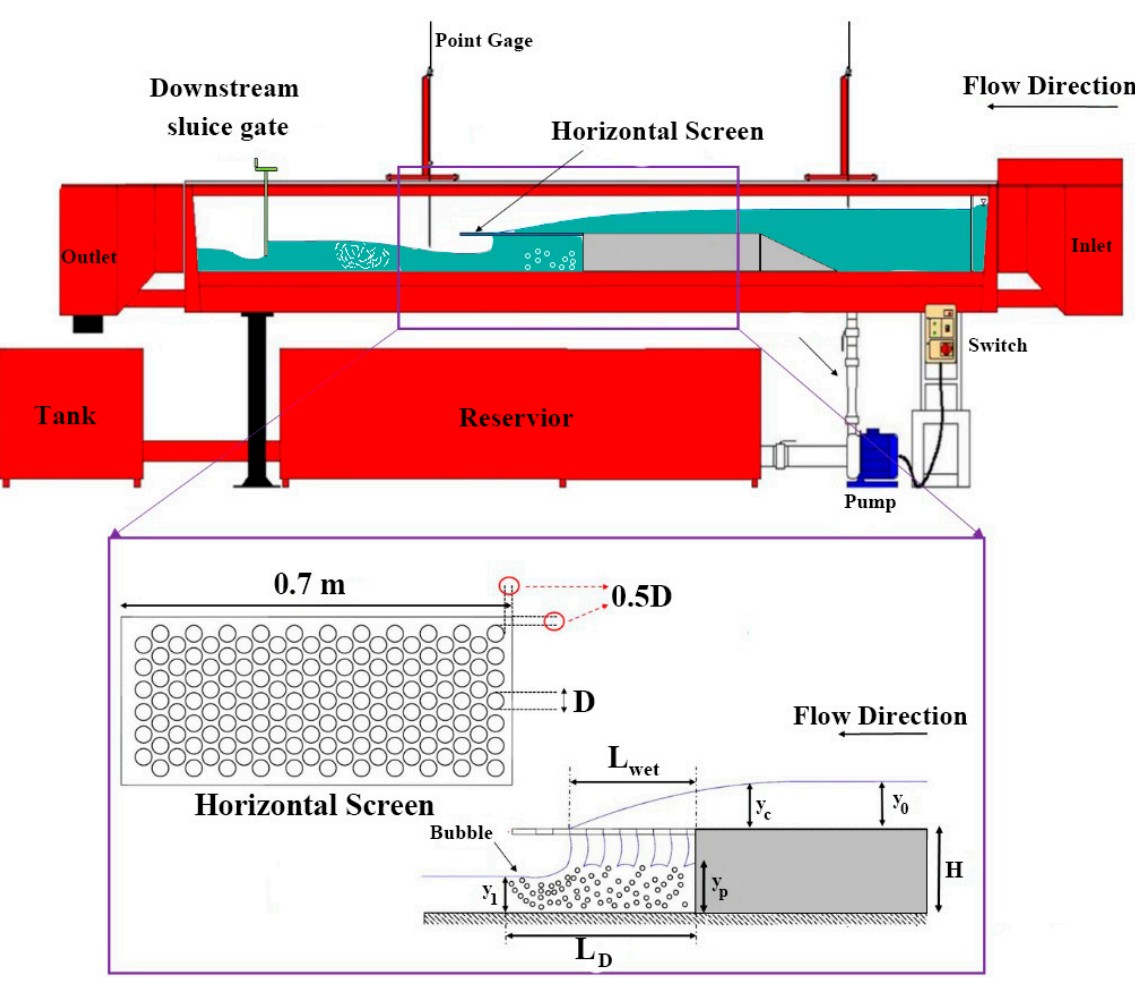

**Figure 1.** Schematic of laboratory flume equipped with horizontal screen.

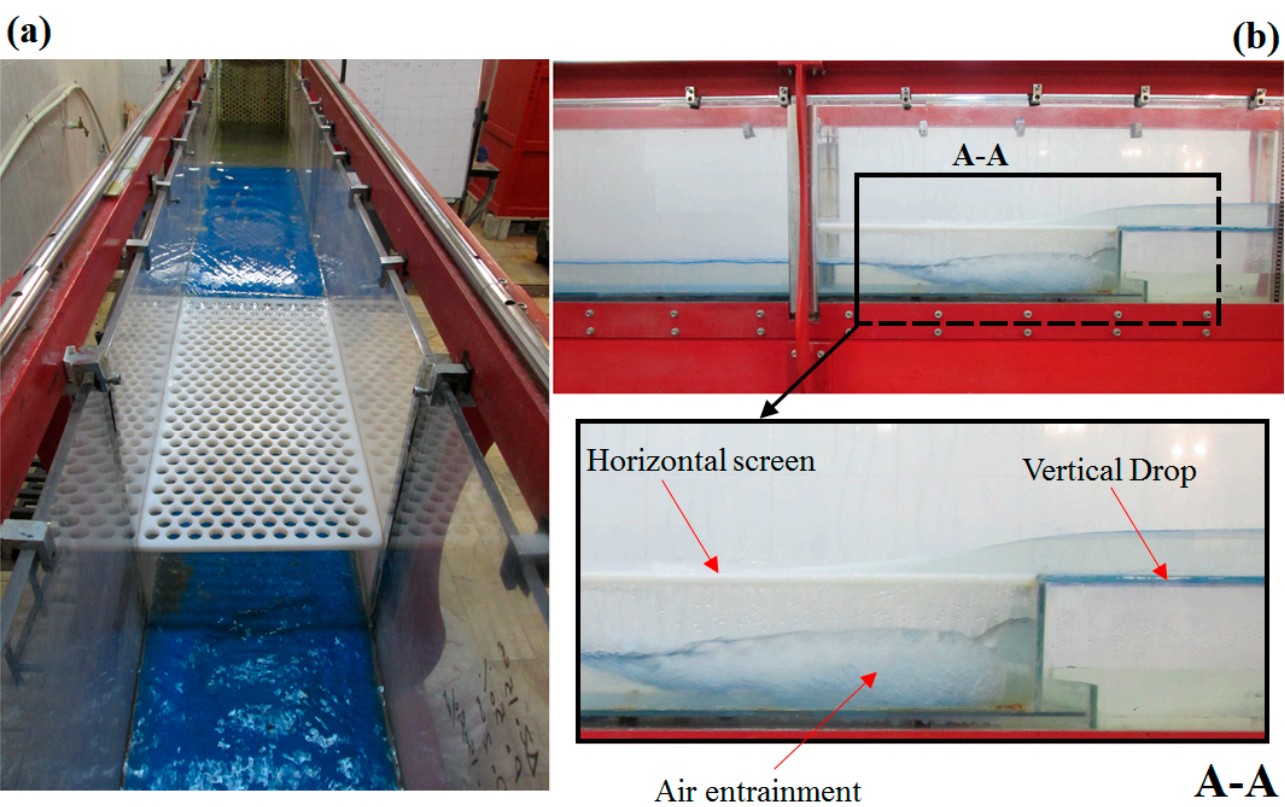

**Figure 2.** Experimental facility of the present study. (**a**) Top view; (**b**) side view.

### 2.3. Support Vector Machine (SVM)

The support vector machine algorithm uses various parameters; one of main parameters is $\gamma$, the correct adjustment of which is very important for improving the predictions. If the data are linear for separation, it tries to select a page with a maximum margin according to Figure 3. If the data are linear, the method selects a page with a maximum margin (as shown in Figure 3). The margin is calculated from Equation (5) and the separating equation is determined from Equation (6) [28].

$$\text{Margin} = \frac{2}{\|w\|} = \frac{2}{w^T w} \tag{5}$$

$$w^T z + b = f(x) = 0 \rightarrow w^T \phi(x) + b = 0 \tag{6}$$

Equation (6) provides a relationship between the dependent and independent variables: $\phi(x)$ is the kernel, $f(x)$ represents the target function, $w$ is a vector coefficient and b is a constant. The SVM algorithm consists of four different kernels, which are presented in Table 2.

**Table 2.** Different kernel functions [24].

| Function | Expression |
|----------|------------|
| Linear | $K(x_i, x_j) = (x_i, x_j)$ |
| Polynomial | $K(x_i, x_j) = \left[(x_i, x_j + 1)\right]^d$ |
| RBF | $K(x_i, x_j) = \exp\left[-\frac{\|x_i - x_j\|^2}{2\sigma^2}\right]$ |
| Sigmoid | $K(x_i, x_j) = \tan h\left[-\alpha(x_i, x_j) + c\right]$ |

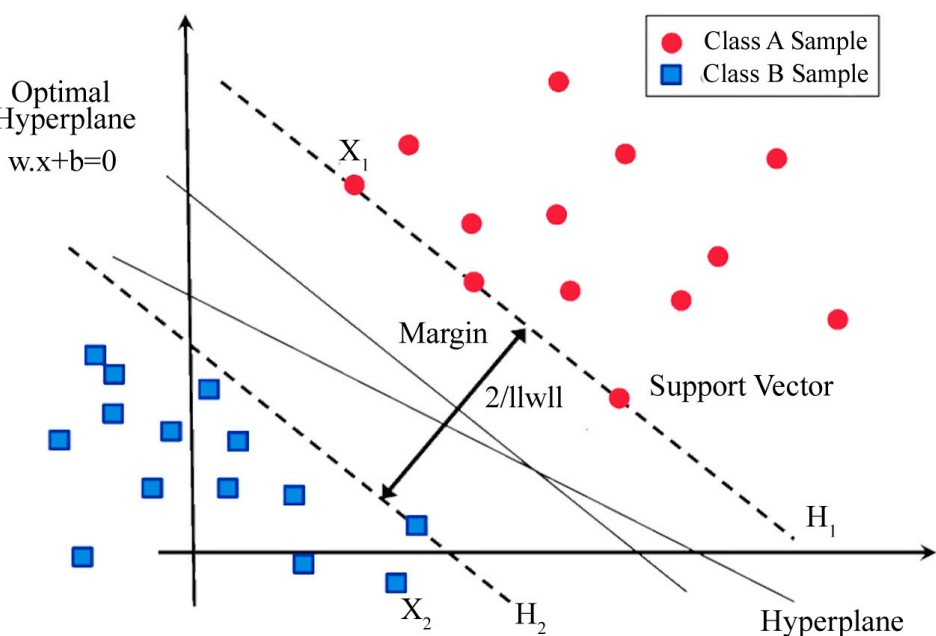

**Figure 3.** Data classification and support vectors.

*2.4. Criteria Evaluation*

In the present study, three evaluation criteria were used to predict the relative energy of the flow and the relative pool depth using the SVM algorithm, which are written as relationships 7, 8 and 9. *RMSE* is the root mean square of the errors, $R^2$ is the coefficient of determination between measured and predicted values, and *KGE* is the Kling–Gupta criterion. In the following equations, the *Pre* subscript corresponds to the predicted values and the *Obs* subscript corresponds to laboratory measurements. It should be noted that the best model is the one in which *RMSE* tends to zero and $R^2$ to one.

$$RMSE = \sqrt{\frac{\sum_{i=1}^{n}\left[\left(\frac{\Delta E}{E_0}, \frac{y_p}{H}\right)_{obs} - \left(\frac{\Delta E}{E_0}, \frac{y_p}{H}\right)_{pre}\right]^2}{n}} \tag{7}$$

$$R^2 = \left[\frac{\sum_{i=1}^{n}\left[\left(\frac{\Delta E}{E_0}, \frac{y_p}{H}\right)_{obs} - \overline{\left(\frac{\Delta E}{E_0}, \frac{y_p}{H}\right)_{obs}}\right] \times \left[\left(\frac{\Delta E}{E_0}, \frac{y_p}{H}\right)_{obs} - \overline{\left(\frac{\Delta E}{E_0}, \frac{y_p}{H}\right)_{pre}}\right]}{\sqrt{\sum_{i=1}^{n}\left(\frac{\Delta E}{E_0}, \frac{y_p}{H}\right)_{obs} - \overline{\left(\frac{\Delta E}{E_0}, \frac{y_p}{H}\right)^2}_{pre}} \times \sqrt{\sum_{i=1}^{n}\left[\left(\frac{\Delta E}{E_0}, \frac{y_p}{H}\right)_{pre} - \overline{\left(\frac{\Delta E}{E_0}, \frac{y_p}{H}\right)^2}_{pre}\right]}}\right]^2 \tag{8}$$

$$KGE = 1 - \sqrt{(R-1)^2 + (\beta-1)^2 + (\gamma-1)^2}$$
$$\beta = \frac{\overline{Pre}}{\overline{Obs}}, \gamma = \frac{CV_{Pre}}{CV_{Obs}} = \frac{\sigma_{Pre}/\overline{Pre}}{\sigma_{Obs}/\overline{Obs}}$$

$$\begin{array}{ll} 0.7 < KGE \leq 1.00 & \text{Verygood} \\ 0.6 < KGE \leq 0.7 & \text{Good} \\ 0.5 < KGE \leq 0.6 & \text{Satisfactory} \\ KGE \leq 0.4 & \text{Unsatisfactory} \end{array} \tag{9}$$

## 3. Results and Discussion

In the present study, the support vector machine method was used to predict the energy dissipation and relative pool depth. Different kernels were used and the RBF function was utilized. To estimate energy dissipation, training and testing modes were used, the results of which are presented in Table 3. A total of 75% of the data was allocated to training and 25% of the data to testing.

**Table 3.** Criteria for evaluating different percentages of training and testing in general (training–testing).

|  | Criteria Evaluation | 60–40% | 70–30% | 75–25% | 80–20% |
|---|---|---|---|---|---|
| First scenario ($\Delta E/E_0$) | RMSE | 0.0429 | 0.0386 | 0.0243 | 0.0395 |
|  | $R^2$ | 0.942 | 0.961 | 0.963 | 0.952 |
| Second scenario ($y_p/H$) | RMSE | 0.0531 | 0.0461 | 0.0328 | 0.0584 |
|  | $R^2$ | 0.952 | 0.966 | 0.972 | 0.95 |

The parameters related to energy dissipation and relative pool depth were identified and seven combinations for relative energy dissipation and three combinations for relative pool depth were introduced, as described in Table 4. The optimal response was calculated using energy dissipation and relative pool depth. To estimate the relative energy dissipation, seven combinations with different inputs for the first scenario and three combinations with different inputs for the second scenario were defined.

**Table 4.** Different input combinations applied in the present study.

| Model | Input Parameters | Model | Input Parameters |
|---|---|---|---|
| First scenario: Relative Energy Dissipation ($\Delta E/E_0$) | | | |
| Model 1 | D/H | Model 5 | $Fr_0$, $y_c/H$ |
| Model 2 | $y_c/H$ | Model 6 | $Fr_0$, D/H |
| Model 3 | $Fr_0$ | Model 7 | $Fr_0$, D/H, $y_c/H$ |
| Model 4 | D/H, $y_c/H$ | | |
| Second scenario: Relative Pool Depth ($y_p/H$) | | | |
| Model 1 | D/H | Model 3 | D/H, $y_c/H$ |
| Model 2 | $y_c/H$ | | |

### 3.1. First Scenario: Relative Energy Dissipation

In the first scenario, a total of seven different combinations was used. The dimensionless parameters were $Fr_0$, D/H and $y_c/H$. These parameters were entered into the support vector machine in the form presented in Table 4, the results of which are presented in Table 5. The combination with the smallest RMSE and the highest coefficient of determination was optimal. According to Figure 4, the results obtained from SVM show that in the first scenario, combination number 3 with input parameter $Fr_0$ had the lowest RMSE, and the highest coefficient of determination and Kling–Gupta value. These values were 0.00565, 0.991 and 0.998, respectively for the training mode. The corresponding results for the testing mode were 0.00489, 0.996 and 0.991, respectively. These settings were selected as the best combination for predicting energy dissipation. To improve visibility, values of RMSE have been multiplied by 100 in the figure.

**Table 5.** The statistical parameters for the first scenario.

|  | Training | | | Testing | | | |
|---|---|---|---|---|---|---|---|
| Model | $R^2$ | RMSE × 100 | KGE | $R^2$ | RMSE × 100 | KGE | $\gamma$ |
| Model 1 | 0.863 | 5.88 | 0.965 | 0.895 | 6.81 | 0.955 | 5 |
| Model 2 | 0.994 | 0.614 | 0.99 | 0.997 | 0.583 | 0.988 | 8 |
| Model 3 | 0.991 | 0.565 | 0.998 | 0.996 | 0.489 | 0.991 | 5 |
| Model 4 | 0.986 | 0.721 | 0.995 | 0.967 | 1.7 | 0.98 | 4 |
| Model 5 | 0.995 | 0.628 | 0.992 | 0.996 | 0.539 | 0.982 | 2 |
| Model 6 | 0.993 | 0.7 | 0.995 | 0.96 | 2.18 | 0.981 | 8 |
| Model 7 | 0.992 | 0.768 | 0.995 | 0.963 | 2.42 | 0.981 | 8 |

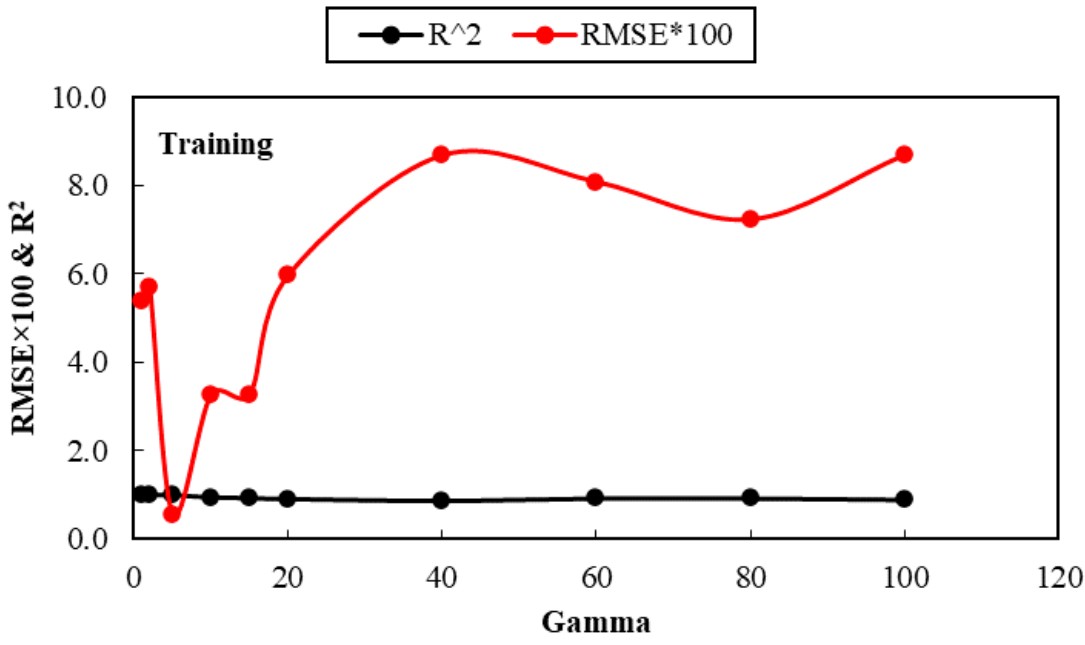

(**a**)

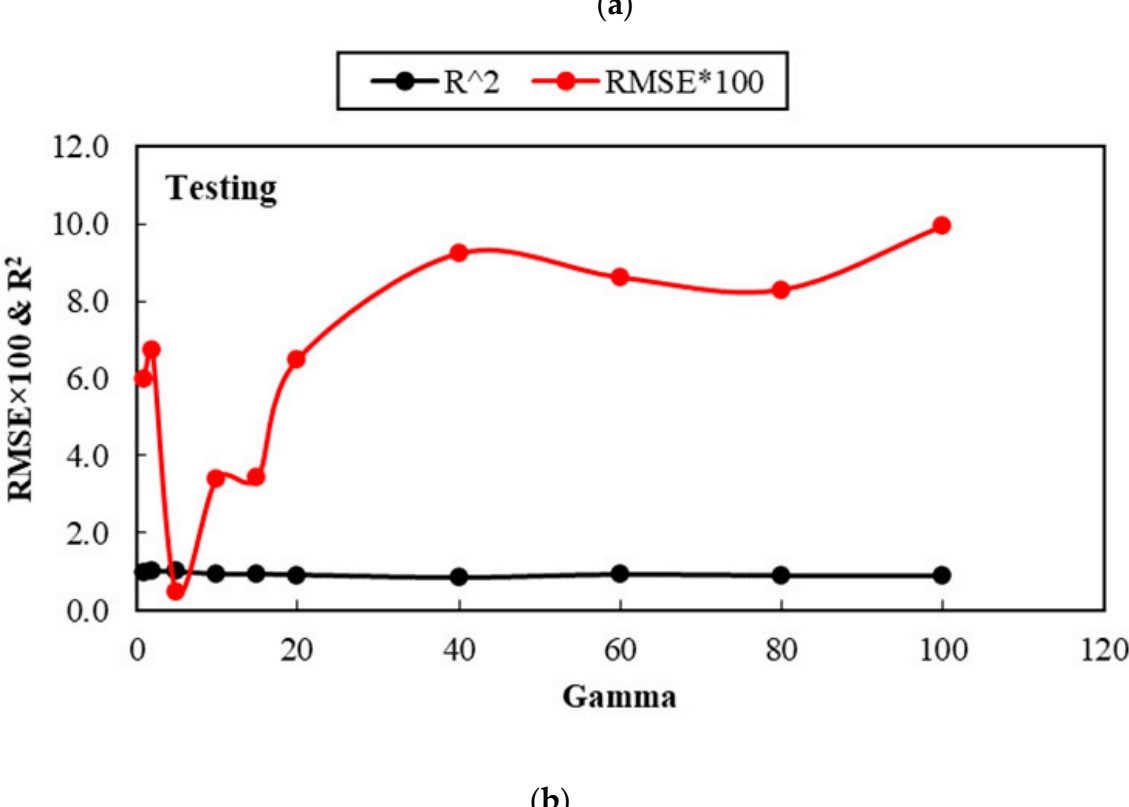

(**b**)

**Figure 4.** Variation of RMSE and $R^2$ vs. Gamma values for best model in the first scenario; (**a**) training, (**b**) testing.

First Scenario SVM Results

Figure 5 shows the experimental and predicted data for the best model in the first scenario.

Figure 5a shows the measured and predicted results for the best composition in the first scenario for the training phase. It was inferred from the figure that the laboratory data were less dispersed than the predicted data, which means that the output data from the SVM were very well matched with a maximum relative error of ±1.67%. Figure 5b compares the laboratory data with the predicted results obtained from the best model in the training phase. The figure shows that there was a very good correlation in this scenario between the laboratory data and the predicted energy consumption. Figure 5c,d show the measured and predicted data for the testing phase; it is seen that the predictions and measurements were in good agreement. They had a relative error of ±1.38%.

The upstream, downstream and relative specific energy dissipations are obtained from Equations (10)–(12), respectively.

$$E_0 = H + \frac{3}{2}y_c \tag{10}$$

$$E_1 = y_1 + \frac{q^2}{2gy_1^2} \tag{11}$$

$$\frac{\Delta E}{E_0} = \frac{E_0 - E_1}{E_0} \tag{12}$$

Figure 6 shows the relative energy dissipation versus the relative critical depth with three relative screen pore diameters. It is inferred from the figure that the values of relative energy dissipation are in good agreement with the results of Hasanniya [29] and indicate that the relative diameter of the pores of the screens has no effect on the relative energy dissipation. For the relative critical depth range of the present study, the range of the downstream Froude number of the simple vertical drop was between 3.5 and 3.9, which is typical for energy dissipation downstream of the simple vertical drop from a Type I stilling basin. Increased turbulence and two-phase mixing of water and air with horizontal screens compared to a stilling basin has reduced the relative energy dissipation. The vertical drop equipped with a horizontal screen has reduced the relative energy dissipation for all three relative diameters of the screens by 31% compared to the type I stilling basin. It is also clear from the graph that the laboratory data are very well matched with the predictions.

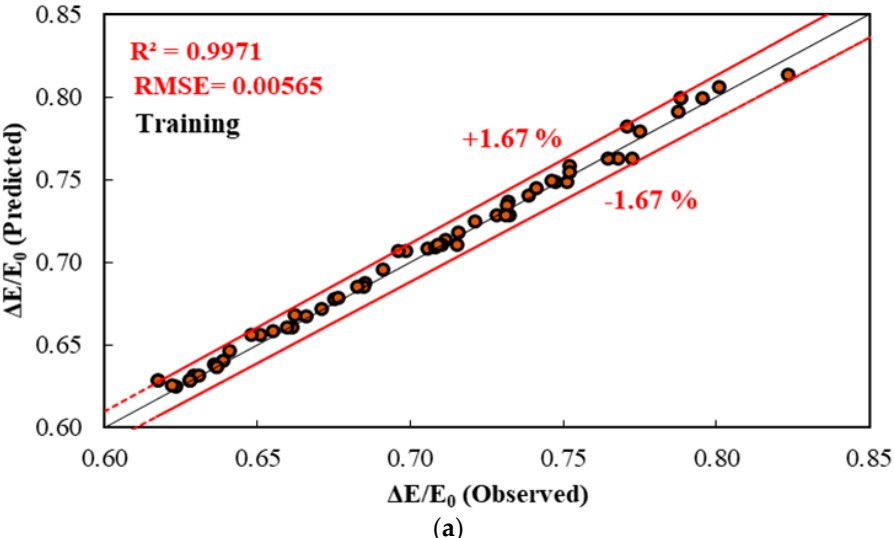

(a)

**Figure 5.** *Cont.*

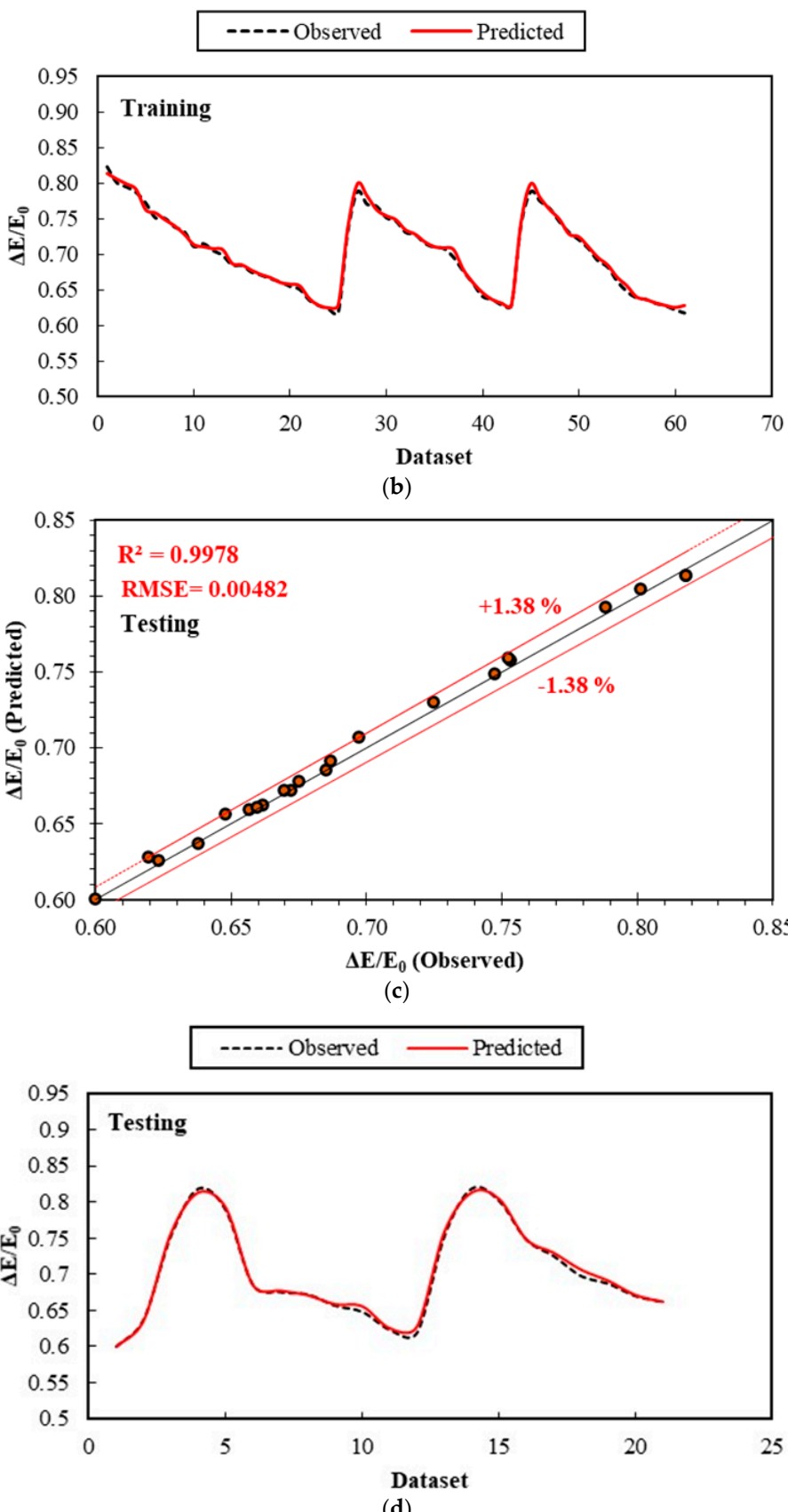

**Figure 5.** Comparison of the dependent and predicted energy dissipation values for the best model (model 3) in the first scenario. (**a**,**b**) Training; (**c**,**d**) testing.

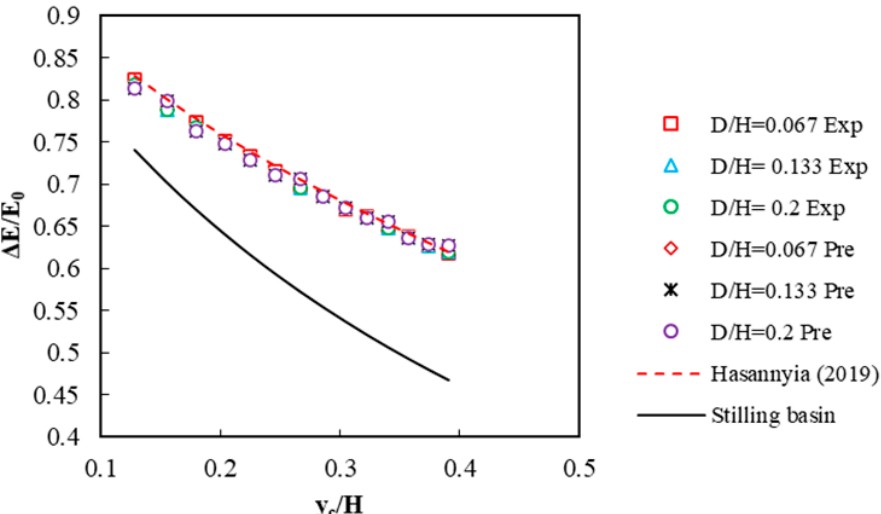

**Figure 6.** Variation of relative energy dissipation versus relative critical depth.

### 3.2. Second Fcenario: Relative Pool Depth

The second scenario consisted of three different combinations that include the dimensionless parameters ($D/H$, $y_c/H$). These parameters were entered into the support vector machine as presented in Table 4, the results of which are presented in Table 6. According to Figure 7, the results obtained from SVM show that in the second scenario, the combination of number 3 with input parameters ($D/H$, $y_c/H$) had the lowest root mean square error, the highest coefficient of determination and the highest Kling–Gupta value, of 0.095, 0.988 and 0.998, respectively for the training mode. The corresponding results were 0.0899, 0.988 and 0.993 for the testing mode, respectively. This combination was selected as the best combination for predicting the relative pool depth. Due to the fact that RMSE values are very small and their changes are not clearly visible in the graph the values have been multiplied by 100 to make the figure more visible.

Second Scenario SVM Results

Figure 8 shows the laboratory data distribution and predicted curves of the best composition for the second scenario.

Figure 8a shows the optimal measured and predicted values for the second scenario for the training phase. It is inferred from the figure that the laboratory data were less dispersed than the predicted data and the output data from the SVM were very well matched with a maximum relative error of ±8.97%. Figure 8b compares the laboratory data with the optimal prediction model in the training phase and shows that there is a very good correlation for this scenario. Figure 8c,d also show the distribution and comparison curves of the laboratory data and the predicted energy dissipation in the testing phase, respectively. It is seen that that the laboratory data corresponded very well with the predicted data, with a maximum relative error of ±8.63%.

**Table 6.** The statistical parameters for the SVM model in the second scenario.

| Model | Training | | | Testing | | | |
|---|---|---|---|---|---|---|---|
| | $R^2$ | RMSE × 100 | KGE | $R^2$ | RMSE × 100 | KGE | $\gamma$ |
| Model 1 | 0.648 | 9.45 | 0.988 | 0.733 | 8.36 | 0.975 | 10 |
| Model 2 | 0.974 | 5.42 | 0.991 | 0.97 | 4.88 | 0.982 | 6 |
| Model 3 | 0.988 | 3.95 | 0.998 | 0.988 | 3.89 | 0.993 | 1 |

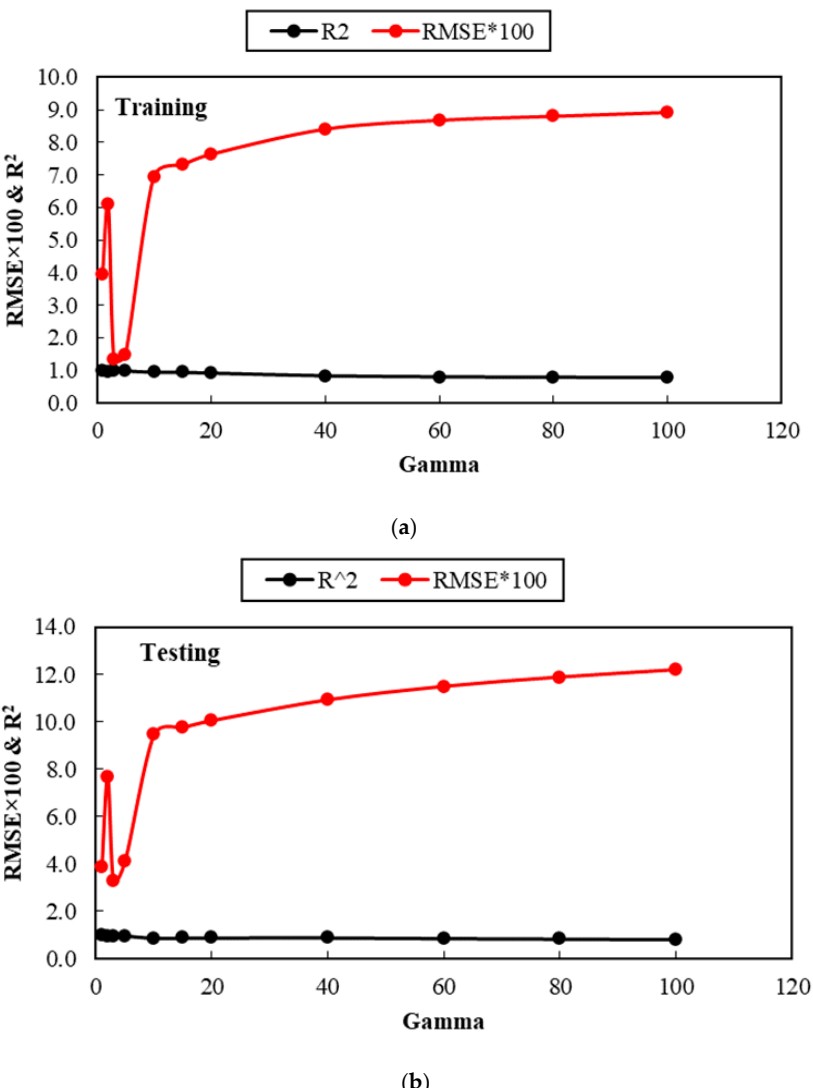

**Figure 7.** Variation of RMSE and $R^2$ vs. Gamma ($\gamma$) values for best model in the second scenario. (**a**) Training; (**b**) testing.

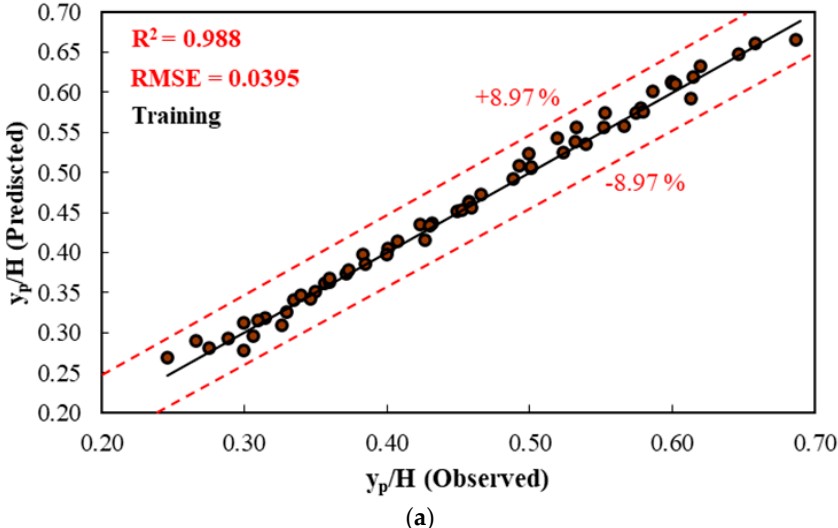

**Figure 8.** *Cont.*

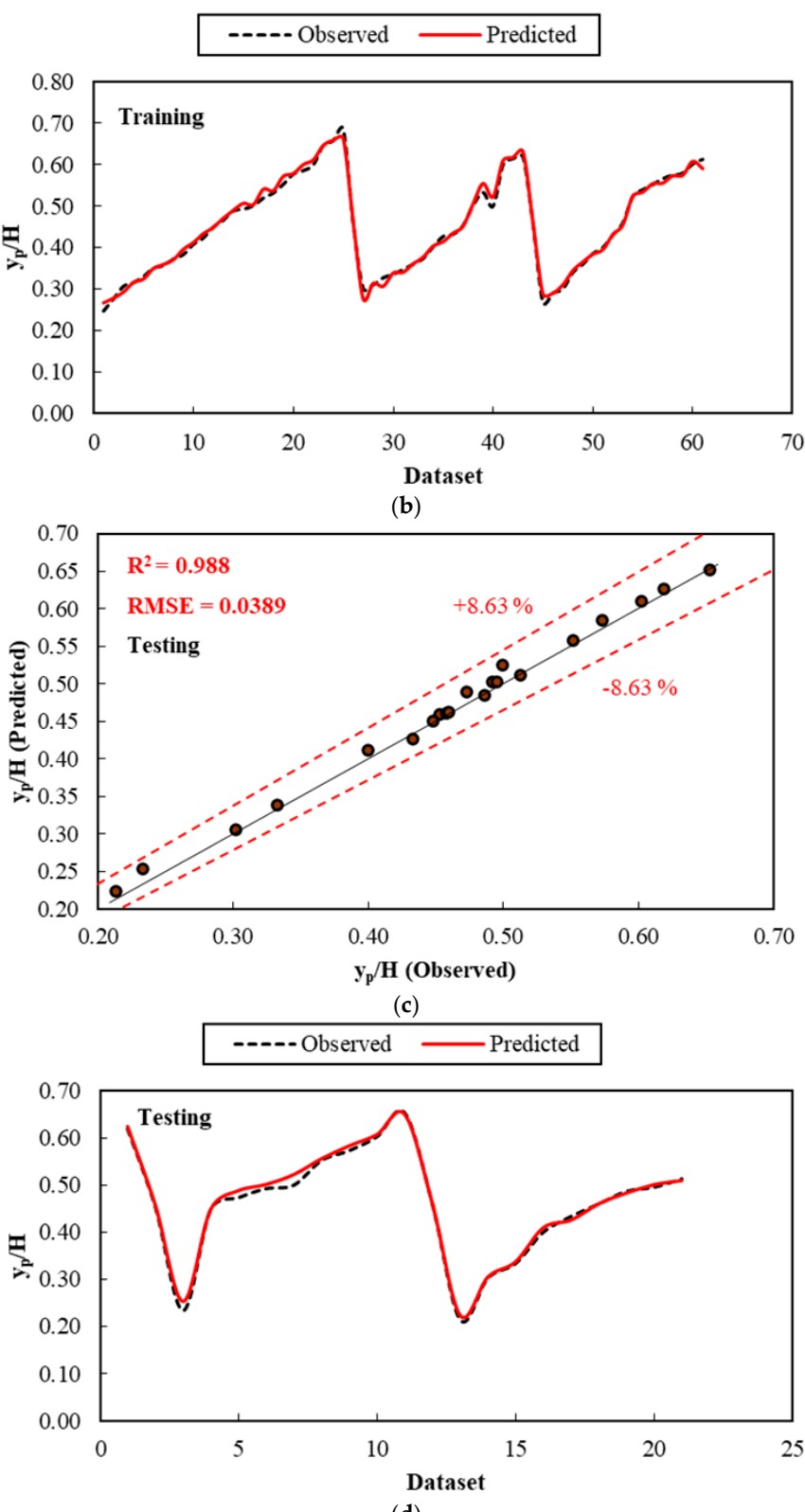

**Figure 8.** Comparison of the observed and predicted energy dissipation values for model 3 in the second scenario. (**a**,**b**) training; (**c**,**d**) testing.

Pool depth values for the three relative screen diameters are shown in Figure 9. It can be seen from the figure that increasing the relative critical depth increased the pool depth for all three relative diameters. Additionally, the depth values of the present research are in good agreement with the results of the Hasanniya [29] study for the same relative diameter. Additionally, by increasing the diameter of the holes in the horizontal screen, the depth of the pool decreased. By increasing the diameters of the screen, the angle of the jet falling from the mesh plate was reduced and this reduced the pool depth; the reduction of air–water interference is another factor that can reduce the depth of the pool. Note that reducing the depth of the pool is an economically effective design for stilling basins.

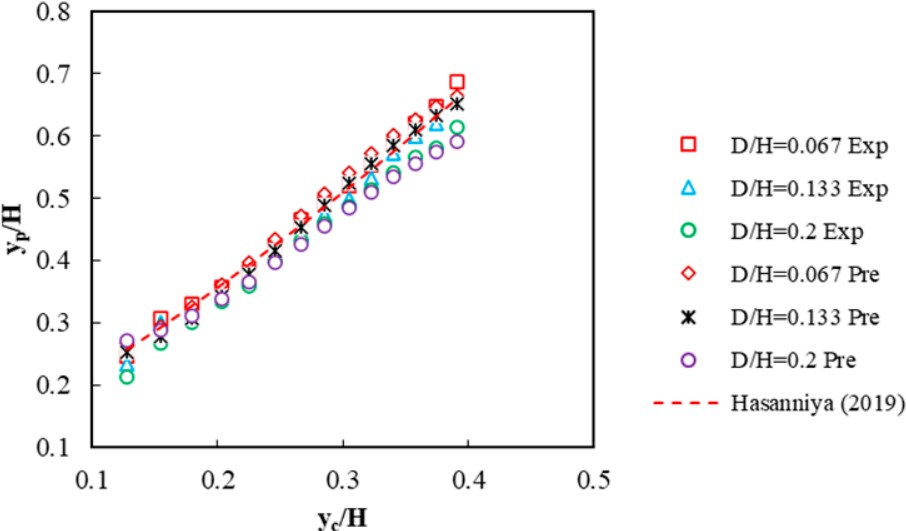

**Figure 9.** Variation of relative pool depth versus relative critical depth.

Compared to the horizontal screen with a relative pore diameter of 0.067, the use of a screen with a relative diameter of 0.2 reduced the relative depth of the pool by 8.5%.

### 3.3. Sensitivity Analysis

A sensitivity analysis was performed to assess the relative importance of input parameters. Sensitivity analysis involves the systematic variation of input parameters and the quantification of their impact, as shown in Table 7.

**Table 7.** Sensitivity Analysis.

| Independent Parameters | Eliminated Parameter | Training | | Testing | |
|---|---|---|---|---|---|
| | | RMSE $\times$ 100 | $R^2$ | RMSE $\times$ 100 | $R^2$ |
| First scenario: Relative energy dissipation | | | | | |
| $Fr_0$, $y_c/H$, $D/H$ | —— | 0.768 | 0.992 | 2.42 | 0.963 |
| $Fr_0$, $y_c/H$ | $D/H$ | 0.628 | 0.995 | 0.539 | 0.996 |
| $Fr_0$, $D/H$ | $y_c/H$ | 0.7 | 0.993 | 2.18 | 0.69 |
| $y_c/H$, $D/H$ | $Fr_0$ | 0.721 | 0.986 | 1.7 | 0.967 |
| Second scenario: Relative pool depth | | | | | |
| $D/H$, $y_c/H$ | —— | 3.95 | 0.988 | 3.89 | 0.988 |
| $D/H$ | $y_c/H$ | 9.45 | 0.648 | 8.36 | 0.733 |
| $y_c/H$ | $D/H$ | 5.42 | 0.974 | 4.88 | 0.97 |

Based on the sensitivity analysis, it was found that in the first scenario, the independent parameter $y_c/H$ and in the second scenario, the independent parameter $D/H$ had the greatest effect on predictions of the relative energy dissipation and relative pool depth. The SVM model has a high sensitivity to the $y_c/H$ parameter for predicting the relative

energy dissipation and a high sensitivity to the *D/H* parameter for predicting the relative pool depth.

## 4. Conclusions

The aim of the present study was to investigate the ability of the support vector machine (SVM) to predict the effect of horizontal screen diameters on hydraulic parameters of vertical drops. A total of 164 experimental datapoints was obtained and three statistical parameters, namely RMSE, $R^2$ and KGE, were used to evaluate the accuracy of the models. The present study used two scenarios, which included the relative energy dissipation ($\Delta E/E_0$) and the relative pool depth ($y_p/H$), which were entered in the support vector machine network as dimensionless parameters. To obtain the best SVM model for parameters $\Delta E/E_0$ and $y_p/H$, input configurations for relative energy dissipation and relative pool depth were introduced into the SVM based models, respectively. The results show that there is a good correlation between the values of $\Delta E/E_0$ and $y_p/H$ obtained by the SVM model and the experimental values of $\Delta E/E_0$ and $y_p/H$ with input parameters. The relative diameter of the screen ($D/H$) and the critical relative depth ($y_c/H$) were found to be the best combination for predicting hydraulic performance. The results of the sensitivity analysis show that the critical relative depth parameter ($y_c/H$) is the most effective parameter for predicting the dependent parameters of the present study.

**Author Contributions:** Conceptualization, R.D., E.A., A.G., J.A. and M.B.; methodology, E.A., A.G. and M.B.; software, E.A. and A.G.; validation, R.D., E.A., A.G., J.A. and M.B.; formal analysis, R.D., E.A., A.G., J.A. and M.B.; resources, R.D. and A.G.; data curation, R.D., E.A., A.G., J.A. and M.B.; writing—original draft preparation, R.D., E.A., A.G., J.A. and M.B.; writing—review and editing, R.D., E.A., A.G., J.A. and M.B.; supervision, R.D., A.G. and J.A.; project administration, R.D., A.G. All authors have read and agreed to the published version of the manuscript.

**Funding:** This research received no external funding.

**Institutional Review Board Statement:** Not applicable.

**Informed Consent Statement:** Not applicable.

**Data Availability Statement:** Data are contained within the article.

**Conflicts of Interest:** The authors declare no conflict of interest.

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
