# Peer review of "SVM Performance for Predicting the Effect of Horizontal Screen Diameters on the Hydraulic Parameters of a Vertical Drop"

_applsci, doi:10.3390/app11094238_

Round 1

Reviewer 1 Report

The present theoretical study is original and important. The application of advanced multivariate statistical approach (support vector machine) contributes significantly to the soundness of the modeling procedure and offers reliable models and prediction options. The sensitivity analysis confirms the correct choice of significant model descriptors.

My recommendation is to shorten the presentation concerning SVM since it is well known and documented method

Author Response

We thank this reviewer for his/her comments and our responses are contained within the attached file.

Reviewer 2 Report

The article is a very pertinent and the Authors have developed an original and interesting topic, full of ideas. Despite this, some integration and clarification are needed: for these reasons I have proposed a "minor revision".

Some comments:

  1. Section 1 “Introduction”: consider a slightly broader description.
  2. Section 4 “Conclusions”: this section is also too concise and schematic: please better focus the conclusions by highlighting the innovative part.

Minor comments:

  • Figure 1 is too little and not clear: please improve dimension and resolution.
  • Figure 2 is too little and not clear: please improve dimension and resolution.
  • Figure 3 is too little and not clear: please improve dimension and resolution.
  • Figures 4 are too little and not clear: please assign a code to every image and improve dimension and resolution.
  • Figures 5 are too little and not clear: please assign a code to every image and improve dimension and resolution.
  • Figure 6 is too little and not clear: please improve dimension and resolution.
  • Figures 7 are too little and not clear: please assign a code to every image and improve dimension and resolution.
  • Figures 8 are too little and not clear: please assign a code to every image and improve dimension and resolution
  • Figure 9 is too little and not clear: please improve dimension and resolution.
  • It is recommended an extensive reading to correct some sentences and typo errors.

Author Response

(The authors gave the same response as above.)

Reviewer 3 Report

The reviewed paper presents a study of using AI technique in prediction of a hydraulic phenomenon. The investigation is interesting and may be published as a new article after a minor revision. The authors are recommended to take into account the following comments and revised the manuscript accordingly.

  1. Please define KGE in the 'Abstract'. It is ambiguous to the reader.
  2. Please add a few sentence about novelty of your study in the 'Abstract', regardless of your results.
  3. Line 29: the sentence looks incorrect, grammatically. Please re-write it.
  4. The channel length (5 m) is relatively short. In this case the flow may not be fully developed at the test section. Did you perform any velocity measurement at the test section? (this study or before)

Author Response

(The authors gave the same response as above.)
